

# GC-MS analysis of phytochemical compounds of *Opuntia megarrhiza* (Cactaceae), an endangered plant of Mexico

Madeleyne Cupido[1], Arturo De-Nova[1,2], María L. Guerrero-González[1], Francisco Javier Pérez-Vázquez[3], Karen Beatriz Méndez-Rodríguez[3] and Pablo Delgado-Sánchez[1]

[1] Facultad de Agronomía y Veterinaria, Universidad Autónoma de San Luis Potosí, Soledad de Graciano Sánchez, San Luis Potosí, Mexico
[2] Instituto de Investigación de Zonas Desérticas, Universidad Autónoma de San Luis Potosí, San Luis Potosí, San Luis Potosí, Mexico
[3] Coordinacion para la Innovación y Aplicación de la Ciencia y la Tecnología, Universidad Autónoma de San Luis Potosí, San Luis Potosí, San Luis Potosí, Mexico

Corresponding authors
Arturo De-Nova,
arturo.denova@uaslp.mx
Pablo Delgado-Sánchez,
pablo.delgado@uaslp.mx

## ABSTRACT

*Opuntia megarrhiza* is an endemic plant used in Mexican traditional medicine for the treatment of bones fractures in humans and domestic animals. One of the most used technique for the detection and characterization of the structure of phytochemical compounds is the Gas Chromatography Coupled to Mass Spectrometry. The goals of the present study were to identify and characterize the phytochemical compounds present in wild individuals of *O. megarrhiza* using this analysis. We used chloroform and methanol extracts from cladodes, and they were analyzed by gas chromatography-electron impact-mass spectrometry. We obtained 53 phytochemical compounds, 19 have been previously identified with some biological activity. Most of these compounds are alkanes, alkenes, aromatic hydrocarbons, fatty acids, and ketones. We detected some fragmentation patterns that are described for the first time for this species. The variety of metabolites presents in *O. megarrhiza* justifies the medicinal use of this plant in traditional medicine and highlight it as a source of phytochemical compounds with potential in medicine and biotechnology.

## INTRODUCTION

The members of Cactaceae represent a diverse evolutionary lineage endemic to America, over 1,450 species belonging to ca. 127 genera (*Barthlott & Hunt, 1993*; *Hunt, Taylor & Charles, 2006*; *Hernández-Hernández et al., 2011*). They are successful plants adapted to arid and semiarid environments, where the conditions imply a constant stress, so they have developed different phytochemical compounds with an important biological activity such as alkaloids, amino acids, antioxidant phenol components (betalains and flavonoids), carotenoids, coumarins, esters, fibers, phytosterols, tannins, terpenes,

tocopherols, and vitamins C and E (*Piattelli, Minale & Prota, 1965*; *Stintzing, Schieber & Carle, 2001*; *Strack, Vogt & Schliemann, 2003*; *Paiz et al., 2010*; *Sim et al., 2010*; *Osorio-Esquivel et al., 2011*; *Harlev et al., 2012*; *Aruwa, Amoo & Kudanga, 2018*; *Araújo et al., 2021*). Bioactive phytochemical compounds are of great interest since their possible applications in biotechnology and industry, and they are usually categorized into phenolic and non-phenolic compounds and pigments (*Martins et al., 2011*; *Aruwa, Amoo & Kudanga, 2018*; *Araújo et al., 2021*). Some of them have nutritional benefits (*Kris-Etherton et al., 2002*; *Kudanga, Nemadziva & Le Roes-Hill, 2017*; *Araújo et al., 2021*; *Yu et al., 2021*), pharmaceutical applications (*Aruwa, Amoo & Kudanga, 2018*; *Araújo et al., 2021*; *Patra et al., 2021*; *Yu et al., 2021*), and are used in the production of nutraceuticals (*Gil-Chávez et al., 2013*; *Aruwa, Amoo & Kudanga, 2018*; *Yu et al., 2021*), in novel food formulations (*Gurrieri et al., 2000*; *Pawar, Killedar & Dhuri, 2017*; *Aruwa, Amoo & Kudanga, 2018*; *Araújo et al., 2021*), and for animal feed supplementation (*Ennouri et al., 2006*; *Aruwa, Amoo & Kudanga, 2018*; *Araújo et al., 2021*).

The species of genus *Opuntia* (Cactaceae) are native of Mexico, where they originated and diversified (*Barthlott & Hunt, 1993*; *Reyes-Agüero, Aguirre & Carlín, 2004*). Different cultures, ancient and modern, have used them as fuel, forage, fences, food, and particularly in traditional medicine (*González-Durán, Riojas & Arreola, 2001*; *Reyes-Agüero, Aguirre-Rivera & Hernández, 2005*; *Andrade-Cetto & Wiedenfeld, 2011*). Several *Opuntia* species have the ability to synthesize molecules with unique and complex structures with therapeutical potential (*Shedbalkar et al., 2010*; *Bargougui, Le Pape & Triki, 2013*; *Weli et al., 2019*). For example, *Opuntia dillenii* (Ker Gawl.) Haw. has beneficial effects for the human health as anti-inflammatory, analgesic, hypoglycemiant, hypocholesterolemiant, and antioxidant (*Perfumi & Tacconi, 1996*; *Park et al., 2001*; *Ghasemzadeh & Ghasemzadeh, 2011*). Other edible species like *Opuntia ficus-indica* (L.) Mill. present antioxidant and antiproliferative activities useful in colon cancer (*Serra et al., 2013*; *Yeddes et al., 2013a*; *Yeddes et al., 2013b*), and have nutraceutical, anticarcinogen, and antiviral properties useful in the digestive processes, reducing the risks of obesity, gastrointestinal suffering and high levels of cholesterol (*Feugang, 2006*; *Bensadón et al., 2010*). Usually, the used parts are the fruit, stem, cladode, and root (*Estrada-Castillón et al., 2012*). It has been estimated that plants have ca. 200,000 different metabolites, primary and secondary (*Pichersky & Gang, 2000*; *Fiehn, 2001*; *Fiehn, 2002*), like aminoacids, fatty acids, carbohydrates, lipids, and more (*Velmurugan & Anand, 2017*; *Banakar & Jayaraj, 2018*). Several studies in *Opuntia* have focused on the analysis of phytochemical compounds as alkaloids, carotenoids, flavonoids, phenols, and vitamins C and E in different plant species in order to discover and produce new drugs for several illnesses (*Yahia & Mondragon-Jacobo, 2011*; *Weli et al., 2019*), as well as nutritional supplements since they provide metabolites, mineral, and vitamins essential for the human organism (*Caballero-Gutiérrez & Gonzáles, 2016*). In the agriculture, some of these compounds are used for the control of phytopathogen microorganisms (*De Corato et al., 2010*). Some other, are used industrially to produce detergents, cosmetics and dermatological products, solvents, lubricants, textiles, and others (*Kim et al., 2019*).

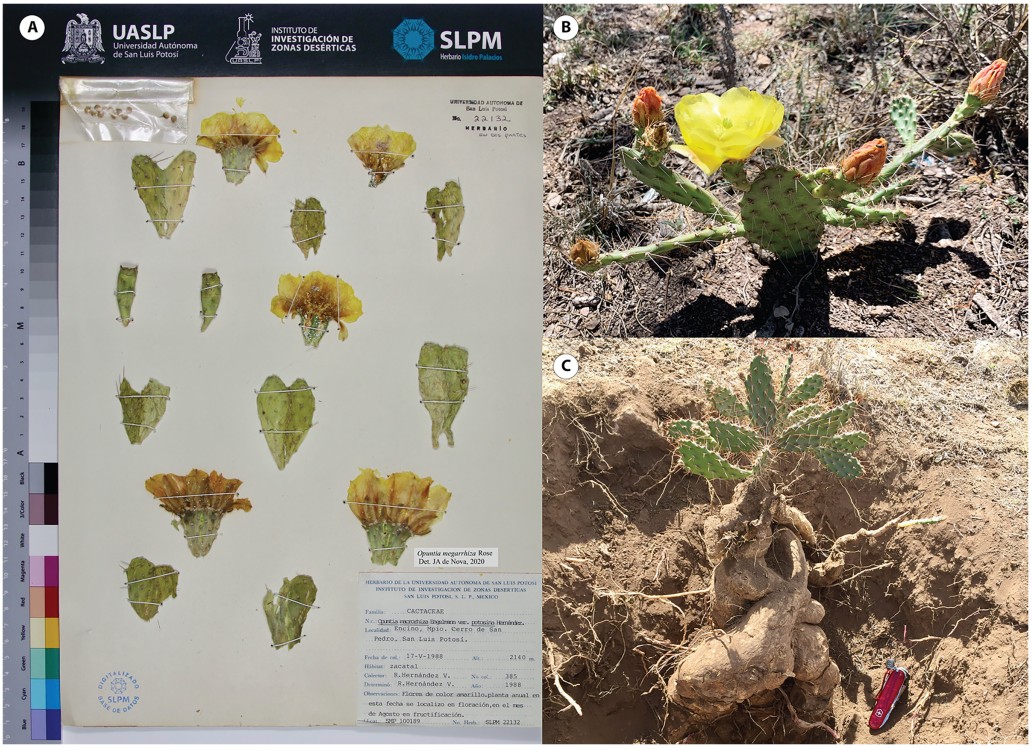

**Figure 1 Opuntia megarrhiza from wild populations.** (A) Herbarium specimen from the studied locality; (B) Flowering adult plant; (C) Adult plant showing part of its characteristic massive roots. Photos (A) and (B) by J.A. De Nova, (C) by P. Delgado.

The main chemical analysis for the detection and characterization of the structure of phytochemical compounds are the Thin-Layer Chromatography, the UV-Vis spectrophotometry, the Nuclear Magnetic Resonance, the liquid chromatography-mass spectrometry (LC-MS), and the Gas Chromatography Coupled to Mass Spectrometric (GC-MS) (*Robertson, 2005*; *Marquet, 2012*). The last one is the most used in metabolomics research since it is a very selective technique for the detection and characterization of metabolites (*Fiehn, 2016*). The LC-MS is a robust technique for general unknown screening, however its major drawback is the lack of universal reference libraries obtained with different instrument types (*Marquet, 2012*), as in GC-MS. The GC-MS together with the metabolomic analysis are a key for the profiling of metabolites in plants since they perform the qualitative and quantitative characterization of all the molecules (metabolites) present in their cellules (*Harrigan & Goodacre, 2012*).

*Opuntia megarrhiza* Rose is a species endemic to Mexico, locally known as "nopal camote" (Fig. 1). It is restricted to some regions of the Chihuahuan Desert, particularly in the State of San Luis Potosi (*Hernández, Gómez-Hinostrosa & Bárcenas, 2001*), and it is listed as endangered in the IUCN Red List (*Hernández et al., 2013*). It grows in different habitats as xerophytic scrubs, oak forest, and other mountain forests (*Hernández, Gómez-Hinostrosa & Bárcenas, 2001*; *Segura-Venegas & Rendón-Aguilar, 2016*). This species is characterized by its massive roots, which are succulent, gross, and deeply buried in ground, 30 to 60 cm long and 5 to 10 cm diameter (*Bravo-Hollis & Sánchez-Mejorada, 1991*;

*Hernández & Godínez, 1994*). The cladodes are relatively small contrasting with other *Opuntia* species. The flowers are yellowish-green to pink, 3 to 5.5 cm long and 2.5 to 6 cm diameter at anthesis. Fruits are ovoid, 2.5 cm long, and the seeds are ca. 4 mm diameter (*Bravo-Hollis & Scheinvar, 1999*; *Hernández, Gómez-Hinostrosa & Bárcenas, 2001*).

*Opuntia megarrhiza* is used by locals in the treatment of bones fractures, both animals and humans (*Hernández, Gómez-Hinostrosa & Bárcenas, 2001*). In Cerro El Borrego (Guadalcazar, San Luis Potosi), the root is applied directly in the fractures, but in other localities like Xoconoxtle (Zaragoza, San Luis Potosi) people use cladodes or the whole plant, and sometimes is mixed with parts of *Cylindropuntia* spp., to make a paste that is applied with bandages in the injury (*Segura-Venegas & Rendón-Aguilar, 2016*). Given the ethnopharmacological value of *O. megarrhiza*, previously highlighted by the empirical traditional medicine, the goals of the present study were to identify and characterize the phytochemical compounds present in cladodes from wild individuals of the species, using GC-MS. There are not previous studies about the bioactive phytochemical compounds in this species, so its phytochemical characterization could contribute to increase the knowledge of the species and its potential biotechnological applications, but also improving bio-valorization and environment.

## MATERIALS AND METHODS

### Sampling

All the protocols involving plants were adhered to relevant ethical guidelines and permissions for plant sampling from herbaria JAAA and SLPM (SGPA/DGGFS/712/0501/18) were used. Non-lethal samples of cladodes were obtained from ten wild individuals, sampling randomly, of *Opuntia megarrhiza* located at Cerro San Pedro (San Luis Potosi). We do not collect whole plants and the identification was conducted in field by taxonomists Arturo De Nova and Eleazar Carranza form herbarium SLPM and registered with photographs then confirmed with Research Grade in iNaturalist (46236747, 46978331). The Fig. 1A show a reference herbarium voucher from the studied locality for identification (SLPM 22132).

### Extracts

The extractions were conducted at Laboratorio de Biotecnologia de la Facultad de Agronomia y Veterinaria, UASLP and Centro Regional de Biociencias, UASLP. Cladode fragments were cleaned using brushes and distillated water to eliminate possible associated microorganisms. We used 95 g of sample, which was macerated to make a semisolid paste, then either 25 ml of methanol (MeOH) for extraction 1 or 25 ml of chloroform ($CHCl_3$) for extraction 2. These solutions were vortex mixed one hour to prevent conglomeration and sedimentation of small particles. The extracts were filtered three times using Whatman paper, grade 1, 5 and 6 in order, in a vacuum chamber. The volumes were adjusted to 10 mg/ml for all extracts. Subsequently, we made a dilution for each solution using acetone as solvent (1:1): (1) acetone-chloroform (($CH_3)_2CO/CHCl_3$), and (2) acetone-methanol (($CH_3)_2CO/MeOH$). Before depositing in a vial, extracts were filtered through a polytetrafluoroethylene (PTFE) or polyvinylidene (PVDF) membranes

(with different hydrophobic adsorption ranges and size exclusion pores), and transferred to a vial for analysis in the GC-MS.

## GC-MS analysis scan mode

This process was conducted at Coordinacion para la Innovacion y la Aplicacion de la Ciencia y la Tecnologia, UASLP. We used the Hewlett Packard gas chromatograph HP 6890, coupled to mass spectrometry detector with electronic impact HP 5973 (Agilent Technologies, Palo Alto, CA, USA). The column exerted was an absorbed silica capillary column of 95% methyl-poly-siloxane and 5% phenyl (HP-5MS; length: 60 m, diameter: 0.25 mm and film 0.25 μm). Helium was used as carrier gas, and the flow rate was 1 ml/min. The GC oven temperature gradient was: 60 °C (hold for 3 min) initially, then increased 5 °C each minute until 300 °C, this final temperature was hold 5 minutes. The transfer line temperature was 280 °C. The ion source temperature was 230 °C and the MS was scanned at 50 to 550 mass range. The essays were processed in the ChemStations software (Houston, TX, USA) to generate the chromatograms for the interpretation of the spectra.

## Identification of phytochemical compounds

The identification was performed by comparing the spectrum of unknown compounds with the spectrum of known compounds in the National Institute of Standards and Technology (NIST98) mass spectral library. Compound name, synonyms, molecular weight, and the mass spectrum for each compound were obtained from NIST Standard Reference 69 and PubChem databases to confirm the GC-MS results. Nomenclature for all compound names was standardized with IUPAC rules. Relative quantification of the compounds present in each sample was obtained from the relative area of the peaks in the chromatograms. Biological activity for each identified compound was obtained with an exhaustive search in scientific publications and from the Dr. Duke phytochemical and ethnobotanical databases. The identity of five compounds showing similarity above 90% (as recommended in *Mangas-Marín et al., 2018*) with phytochemical compounds previously reported with biological activity was verified by using the commercial pure standards: The used standards were 1,3-benzothiazole (Purity: minimum 97.0%), henicosane (Purity: minimum 99.5%), hentriacontane (Purity: minimum 98.0%), methyl hexadecanoate (Purity: minimum 98.5%), triacontane (Purity: minimum 98.0%), were purchased from Sigma (St. Louis, MO, USA). All standards were diluted in acetone as the final solvent at a concentration of 5 ppm and were analyzed in the GC-MS using the same parameters than in the samples.

## RESULTS

A total of 53 phytochemical compounds were detected based on the analyses of the obtained chromatograms (Tables 1–4). The $(CH_3)_2CO/MeOH$ extract showed 11 peaks with the PVDF membrane filter and 12 with the PTFE. The $(CH_3)_2CO/CHCl_3$ extract showed 23 peaks with PVDF and seven with PTFE (Fig. 2).

**Table 1 Phytochemical compounds identified by GC-MS from MeOH extract with PVDF membrane filter.**

| Peak No. | RT (min) | Name of the compound | Molecular weight (g/mol) | Peak area (%) | Similarity* (%) | Molecular formula | Compound nature |
|---|---|---|---|---|---|---|---|
| 1 | 8.21 | 2,3,6,7-tetramethyloctane | 170.33 | 1.17 | 64 | $C_{12}H_{26}$ | Alkane |
| 2 | 8.33 | 2,3,5,8-tetramethyldecane | 198.38 | 1.78 | 64 | $C_{14}H_{30}$ | Alkane |
| 3 | 15.35 | 1,3-ditert-butylbenzene | 190.32 | 4.19 | 95 | $C_{14}H_{22}$ | Aromatic Hydrocarbon |
| 4 | 16.74 | methyl (2R)-5-oxo-2-propan-2-ylhexanoate | 186.25 | 2.78 | 53 | $C_{10}H_{18}O$ | Ester |
| 5 | 17.20 | 4-propan-2-ylcyclohexane-1,3-dione | 154.21 | 2.19 | 60 | $C_9H_{14}O_2$ | Ketone |
| 6 | 17.28 | heptadecane | 240.46 | 1.03 | 59 | $C_{17}H_{36}$ | Alkane |
| 7 | 21.25 | 7-methylhexadecane | 240.46 | 1.00 | 72 | $C_{17}H_{36}$ | Alkane |
| 8 | 22.69 | 2,3,6-trimethyldecane | 184.36 | 1.80 | 64 | $C_{13}H_{28}$ | Alkane |
| 9 | 31.75 | hexadecanoic acid | 256.42 | 4.67 | 97 | $C_{16}H_{32}O_2$ | Fatty acid |
| 10 | 35.43 | octadecanoic acid | 284.47 | 2.47 | 93 | $C_{18}H_{36}O_2$ | Fatty acid |
| 11 | 46.72 | 1,54-dibromotetrapentacontane | 917.2 | 1.41 | 76 | $C_{54}H_{108}Br_2$ | Halogenated hydrocarborn |

Notes:
* Percentage of similarity to the reference spectrum of the NIST library.
RT, retention time.

**Table 2 Phytochemical compounds identified by GC-MS from MeOH extract with PTFE membrane filter.**

| Peak No. | RT (min) | Name of the compound | Molecular weight (g/mol) | Peak area (%) | Similarity* (%) | Molecular formula | Compound nature |
|---|---|---|---|---|---|---|---|
| 1 | 8.33 | 4-methyldecane | 156.30 | 1.44 | 64 | $C_{11}H_{24}$ | Alkane |
| 2 | 15.35 | 1,3-ditert-butylbenzene | 190.32 | 3.53 | 95 | $C_{14}H_{22}$ | Aromatic Hydrocarbon |
| 3 | 17.20 | 4-propan-2-ylcyclohexane-1,3-dione | 154.21 | 1.84 | 53 | $C_9H_{14}O_2$ | Ketone |
| 4 | 21.61 | heptacosane | 380.73 | 0.98 | 83 | $C_{27}H_{56}$ | Alkane |
| 5 | 22.05 | 2,4-ditert-butylphenol | 206.32 | 2.13 | 97 | $C_{14}H_{22}O$ | Aromatic Hydrocarbon |
| 6 | 22.69 | pentacosane | 352.68 | 1.66 | 64 | $C_{25}H_{52}$ | Alkane |
| 7 | 32.89 | henicosane | 296.57 | 0.98 | 90 | $C_{21}H_{44}$ | Alkane |
| 8 | 46.73 | nonacosane | 408.8 | 2.28 | 95 | $C_{29}H_{60}$ | Alkane |
| 9 | 48.04 | triacontane | 422.81 | 1.4 | 96 | $C_{30}H_{62}$ | Alkane |
| 10 | 49.33 | hentriacontane | 436.83 | 1.63 | 93 | $C_{31}H_{64}$ | Alkane |
| 11 | 50.57 | octacosane | 394.76 | 1.28 | 95 | $C_{28}H_{58}$ | Alkane |
| 12 | 52.22 | (3S,8S,9S,10R,13R,14S,17R)-17-[(2R,5S)-5-ethyl-6-methylheptan-2-yl]-10,13-dimethyl-2,3,4,7,8,9,11,12,14,15,16,17-dodecahydro-1H-cyclopenta[a]phenanthren-3-ol | 414.70 | 1.76 | 90 | $C_{29}H_{50}O$ | Lipids |

Notes:
* Percentage of similarity to the reference spectrum of the NIST library.
RT, retention time.

The analysis from $(CH_3)_2CO$/MeOH extract with PVDF membrane filter showed the presence of 11 phytochemical constituents. Five alkanes: 2,3,6,7-tetramethyloctane (1.17%), 2,3,5,8-tetramethyldecane (1.78%), heptadecane (1.03%), 7-methylhexadecane (1%), 2,3,6-trimethyldecane (1.80%). One aromatic hydrocarbon: 1,3-ditert-butylbenzene (4.19%). One ester: methyl (2R)-5-oxo-2-propan-2-ylhexanoate (2.78%). One ketone: 4-propan-2-ylcyclohexane-1,3-dione (2.19%). One Halogenated hydrocarbons: 1,54-dibromotetrapentacontane (1.41%). Two fatty acids: hexadecanoic acid (4.67%), octadecanoic acid (2.47%) (Fig. 2A, Table 1).

**Table 3 Phytochemical compounds identified by GC-MS from $CHCl_3$ extract with PVDF membrane filter.**

| Peak No. | RT (min) | Name of the compound | Molecular weight (g/mol) | Peak area (%) | Similarity[*] (%) | Molecular formula | Compound nature |
|---|---|---|---|---|---|---|---|
| 1 | 10.45 | 4-ethyl-1,2-dimethylbenzene | 134.21 | 2.27 | 93 | $C_{10}H_{14}$ | Aromatic Hydrocarbon |
| 2 | 11.34 | 1,2,4,5-tetramethylbenzene | 134.21 | 0.51 | 81 | $C_{10}H_{14}$ | Aromatic Hydrocarbon |
| 3 | 11.47 | 1,2,3,4-tetramethyl-5-methylidenecyclopenta-1,3-diene | 134.21 | 0.82 | 95 | $C_{10}H_{14}$ | Alkene |
| 4 | 14.53 | 1,3-benzothiazole | 135.18 | 5.33 | 95 | $C_7H_5NS$ | Aromatic Hydrocarbon |
| 5 | 18.59 | (E,7R,11R)-3,7,11,15-tetramethylhexadec-2-en-1-ol | 296.53 | 0.45 | 42 | $C_{20}H_{40}O$ | Alcohol |
| 6 | 22.70 | 6-hexyloxan-2-one | 184.27 | 0.58 | 59 | $C_{11}H_{20}O_2$ | Ketone |
| 7 | 30.09 | (E)-octadec-5-ene | 252.47 | 0.67 | 78 | $C_{18}H_3$ | Alkene |
| 8 | 30.95 | 7,9-di*tert*-butyl-1-oxaspiro[4.5]deca-6,9-diene-2,8-dione | 276.37 | 1.61 | 50 | $C_{17}H_{24}O_3$ | Ketone |
| 9 | 31.01 | methyl hexadecanoate | 270.45 | 1.1 | 93 | $C_{17}H_{34}O_2$ | Fatty acid |
| 10 | 33.98 | (E)-octadec-9-ene | 252.5 | 1.65 | 89 | $C_{18}H_{36}$ | Alkene |
| 11 | 34.20 | methyl (9Z,12Z)-octadeca-9,12-dienoate | 294.47 | 0.65 | 96 | $C_{19}H_{34}O_2$ | Fatty acid |
| 12 | 34.31 | (3Z,13Z)-2-methyloctadeca-3,13-dien-1-ol | 280.5 | 0.45 | 53 | $C_{19}H_{36}O$ | Alcohol |
| 13 | 34.78 | methyl octadecanoate | 298.50 | 0.97 | 93 | $C_{19}H_{38}O_2$ | Fatty acid |
| 14 | 34.88 | tridecanedial | 212.33 | 0.58 | 62 | $C_{13}H_{24}O_2$ | Aldehyde |
| 15 | 34.97 | 1-(7-hydroxy-8,9-dimethoxy-17-oxa-5,15-diazahexacyclo[13.4.3.0$^{1,16}$.0$^{4,12}$.0$^{6,11}$.0$^{12,16}$]docosa-6,8,10-trien-5-yl)ethanone | 414.5 | 0.84 | 52 | $C_{23}H_{30}N_2O_5$ | Ketone |
| 16 | 35.84 | butyl hexadecanoate | 312.53 | 4.83 | 87 | $C_{20}H_{40}O_2$ | Fatty acid |
| 17 | 36.24 | dioctadecoxy(oxo)phosphanium | 585.9 | 0.55 | 53 | $C_{36}H_{74}O_3P^+$ | Fatty acid |
| 18 | 39.21 | 2-methylpropyl octadecanoate | 340.58 | 3.10 | 87 | $C_{22}H_{44}O_2$ | Fatty acid |
| 19 | 39.39 | tetratriacontane | 478.91 | 0.92 | 90 | $C_{34}H_{70}$ | Alkane |
| 20 | 40.96 | tetrapentacontane | 759.45 | 0.93 | 80 | $C_{54}H_{110}$ | Alkane |
| 21 | 42.48 | icosane | 282 | 1.34 | 68 | $C_{20}H_{42}$ | Alkane |
| 22 | 45.81 | (6E,10E,14E,18E)-2,6,10,14,18-pentamethylicosa-2,6,10,14,18-pentaene | 350.6 | 4.80 | 74 | $C_{25}H_{50}$ | Alkene |
| 23 | 52.21 | 17-(5-ethyl-6-methylheptan-2-yl)-10,13-dimethyl-2,7,8,9,11,12,14,15,16,17-decahydro-1*H*-cyclopenta[a]phenanthrene | 396.7 | 13.80 | 60 | $C_{29}H_{48}$ | Alkene |

**Notes:**
[*] Percentage of similarity to the reference spectrum of the NIST library.
RT, retention time.

The analysis from the $(CH_3)_2CO$/MeOH extract with PTFE membrane filter detected 12 phytochemical constituents. Eight alkanes: 4-methyldecane (1.44%), heptacosane (0.98%), pentacosane (1.66%), henicosane (0.98%), nonacosane (2.28%), triacontane (1.4%), hentriacontane (1.63%), octacosane (1.28%). Two aromatic hydrocarbons: 1,3-di*tert*-butylbenzene (3.53%), 2,4-di*tert*-butylphenol (2.13%). One ketone: 4-propan-2-ylcyclohexane-1,3-dione (1.84%). One lipid: (3S,8S,9S, 10R,13R, 14S,17R)-17-[(2R,5S)-5-ethyl-6-methylheptan-2-yl]-10,13-dimethyl-2,3,4,7,8,9,11,12, 14,15,16,17-dodecahydro-1*H*-cyclopenta[a]phenanthren-3-ol (1.76%) (Fig. 2B, Table 2).

**Table 4 Phytochemical compounds identified by GC-MS from CHCl₃ extract with PTFE membrane filter.**

| Peak No. | RT (min) | Name of the compound | Molecular Weight (g/mol) | Peak area (%) | Similarity* (%) | Molecular formula | Compound nature |
|---|---|---|---|---|---|---|---|
| 1 | 10.25 | 1-ethyl-2,4-dimethylbenzene | 134.21 | 2.51 | 64 | $C_{10}H_{14}$ | Aromatic Hydrocarbon |
| 2 | 10.45 | 4-ethyl-1,2-dimethylbenzene | 134.21 | 4.24 | 80 | $C_{10}H_{14}$ | Aromatic Hydrocarbon |
| 3 | 11.48 | 2-ethyl-1,3-dimethylbenzene | 134.21 | 0.83 | 74 | $C_{10}H_{14}$ | Aromatic Hydrocarbon |
| 4 | 22.07 | 2,4-di*tert*-butylphenol | 206.32 | 5.28 | 83 | $C_{14}H_{22}O$ | Aromatic Hydrocarbon |
| 5 | 36.26 | 1-(6-methylheptan-2-yl)-4-(4-methylpentyl)cyclohexane | 280.53 | 1.20 | 53 | $C_{20}H_{40}$ | Alkane |
| 6 | 39.22 | 2-methylpropyl octadecanoate | 340.58 | 10.83 | 90 | $C_{22}H_{44}O_2$ | Fatty acid |
| 7 | 41.77 | 2-(2-ethylhexoxycarbonyl)benzoic acid | 278.34 | 7.55 | 53 | $C_{16}H_{22}O_4$ | Ester |

Notes:
* Percentage of similarity to the reference spectrum of the NIST library.
RT, retention time.

The analysis from the $(CH_3)_2CO/CHCl_3$ extract with PVDF membrane filter showed the presence of 23 phytochemical constituents. Three alkanes: tetratriacontane (0.92%), tetrapentacontane (0.93%), icosane (1.34%). Three aromatic hydrocarbons: 4-ethyl-1,2-dimethylbenzene (2.27%), 1,2,4,5-tetramethylbenzene (0.51%), 1,3-benzothiazole (5.33%). Three ketones: 6-hexyloxan-2-one (0.58%), 7,9-di*tert*-butyl-1-oxaspiro[4.5]deca-6,9-diene-2,8-dione (1.61%), 1-(7-hydroxy-8,9-dimethoxy-17-oxa-5,15-diazahexacyclo [13.4.3.0$^{1,16}$.0$^{4,12}$.0$^{6,11}$.0$^{12,16}$]docosa-6,8,10-trien-5-yl)ethanone (0.84%). Two alcohols: (*E*,7*R*,11*R*)-3,7,11,15-tetramethylhexadec-2-en-1-ol (0.45%), (3*Z*,13*Z*)-2-methyloctadeca-3,13-dien-1-ol (0.45%). One aldehyde: tridecanedial (0.58%). Five alkenes: 1,2,3,4-tetramethyl-5-methylidenecyclopenta-1,3-diene (0.82%), (*E*)-octadec-5-ene (0.67%), (*E*)-octadec-9-ene (1.65%), (6*E*,10*E*,14*E*,18*E*)-2,6,10,14,18-pentamethylicosa-2,6,10,14,18-pentaene (4.80%), 17-(5-ethyl-6-methylheptan-2-yl)-10,13-dimethyl-2,7,8,9,11,12, 14,15,16,17-decahydro-1*H*-cyclopenta[a]phenanthrene (13.80%). Six fatty acids: methyl hexadecanoate (1.1%), methyl (9*Z*,12*Z*)-octadeca-9,12-dienoate (0.65%), methyl octadecanoate (0.97%), butyl hexadecanoate (4.83%), dioctadecoxy(oxo)phosphanium (0.55%), 2-methylpropyl octadecanoate (3.10%) (Fig. 2C, Table 3).

The analysis from $(CH_3)_2CO/CHCl_3$ extract with PTFE membrane filter seven compounds were observed. One alkane: 1-(6-methylheptan-2-yl)-4-(4-methylpentyl) cyclohexane (1.20%). Four aromatic hydrocarbons: 1-ethyl-2,4-dimethylbenzene (2.51%), 4-ethyl-1,2-dimethylbenzene (4.24%), 2-ethyl-1,3-dimethylbenzene (0.83%), 2,4-di*tert*-butylphenol (5.28%). One ester: 2-(2-ethylhexoxycarbonyl)benzoic acid (7.55%). One fatty acid: 2-methylpropyl octadecanoate (10.83%) (Fig. 2D, Table 4).

The analyses reveled different nature kinds for the identified compounds such as alkanes, aromatic hydrocarbons, esters, ketones halogenated hydrocarbons, alcohols, aldehydes, alkenes, lipids, and fatty acids, some of them with a biological activity previously reported (Tables 5–7). From the identified compounds, 19 shown similarities to
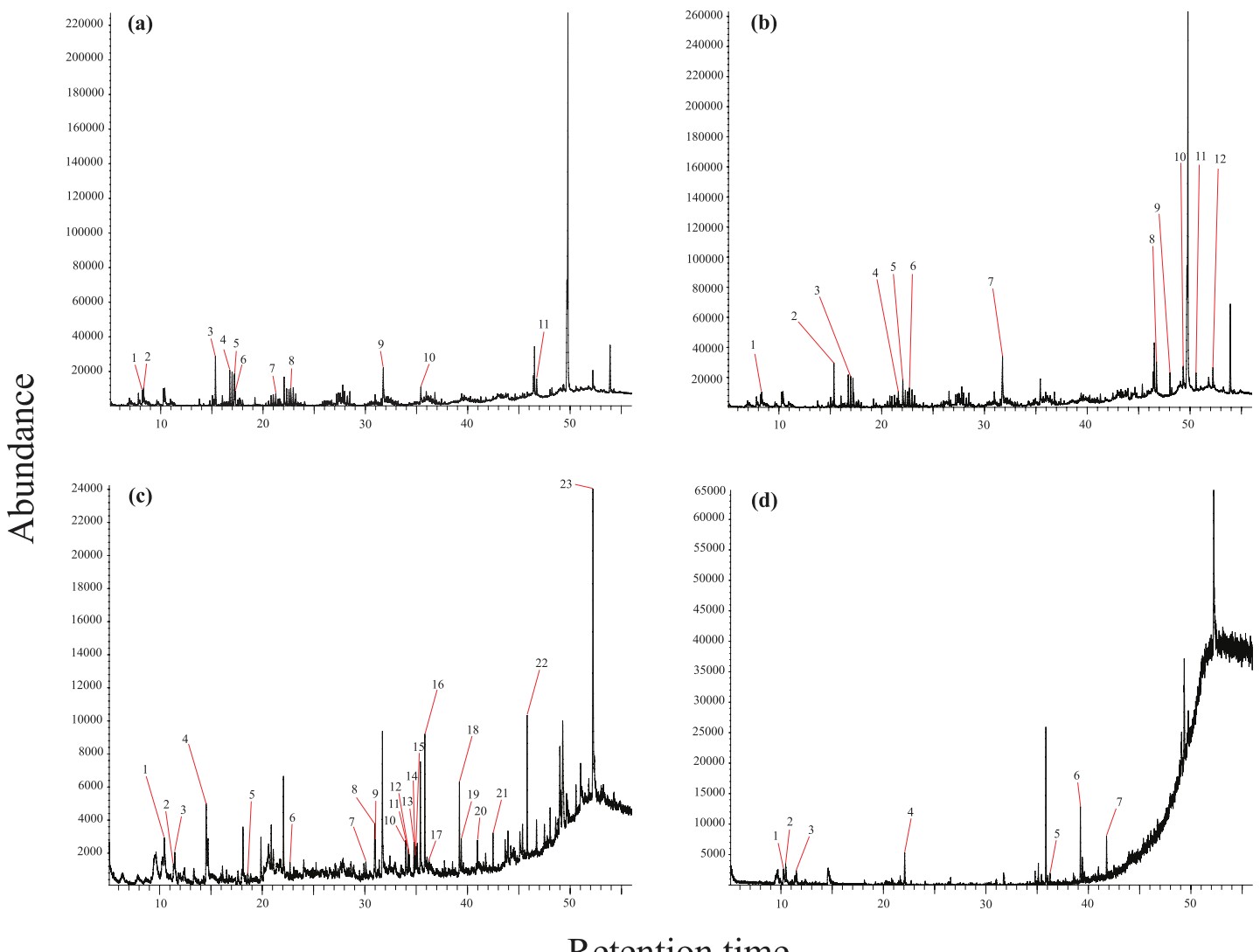

**Figure 2 GC-MS chromatograms.** (A) MeOH extract with PVDF membrane filter, (B) MeOH extract with PTFE membrane filter, (C) CHCl₃ extract with PVDF membrane filter, and (D) CHCl₃ extract with PTFE membrane filter.

**Table 5 Number and type of phytochemical compounds in *Opuntia megarrhiza* by extract and membrane filter.**

| Extract/ Filter | A | Ah | Fa | K | Ak | Al | E | Ad | L | Hh | Total |
|---|---|---|---|---|---|---|---|---|---|---|---|
| MeOH/PVDF | 5 | 1 | 2 | 1 | 0 | 0 | 1 | 0 | 0 | 1 | 11 |
| MeOH/PTFE | 8 | 2 | 0 | 1 | 0 | 0 | 0 | 0 | 1 | 0 | 12 |
| CHCl₃/PVDF | 3 | 3 | 6 | 3 | 5 | 2 | 0 | 1 | 0 | 0 | 23 |
| CHCl₃/PTFE | 1 | 4 | 1 | 0 | 0 | 0 | 1 | 0 | 0 | 0 | 7 |
| Total | 17 | 10 | 9 | 5 | 5 | 2 | 2 | 1 | 1 | 1 | 53 |

**Note:**
Alkanes (A), Aromatic hydrocarbons (Ah), Esters (E), Ketones (K), Halogenated hydrocarbons (Hh), Alcohols (Al), Aldehydes (Ad), Alkenes (Ak), Lipids (L), Fatty acids (Fa).

**Table 6 Phytochemical compounds with biological activity identified by GC-MS in MeOH extract.**

| Peak No. | Name of the compound | Molecular weight (g/mol) | Peak area (%) | Molecular formula | Ions (*m/z*) | Compound nature | Biological activity |
|---|---|---|---|---|---|---|---|
| PVDF filter | | | | | | | |
| 1 | heptadecane | 240.46 | 1.03 | $C_{17}H_{36}$ | 57, 71, 85 | Alkane | Antifungal (*Adeleye, Daniels & Omadime, 2010*), antimicrobial (*Rahbar, Shafagha & Salimi, 2012*), anti-inflammatory and antioxidative (*Kim et al., 2013*) |
| 2 | hexadecanoic acid | 256.42 | 4.67 | $C_{16}H_{32}O_2$ | 57, 73, 129 | Fatty acid | Anti-inflammatory (*Aparna et al., 2012*) antialopecic, anti-androgenic, antifibrinolytic, antioxidant, antipsychotic, hemolytic, hypocholesterolemic, nematicide, pesticide and 5-Alpha reductase inhibitor (*USDA, 1992*–2016 [U.S. Department of Agriculture, Agricultural Research Service]). Dr. Duke's) |
| 3 | octadecanoic acid | 284.47 | 2.14 | $C_{18}H_{36}O_2$ | 73, 55, 129 | Fatty acid | Antibacterial, antifungal and antitumoral (*Gehan et al., 2009*; *Hsouna et al., 2011*) |
| PTFE filter | | | | | | | |
| 1 | heptacosane | 380.73 | 0.98 | $C_{27}H_{56}$ | 57, 71, 85 | Alkane | Antioxidant (*Marrufo et al., 2013*) |
| 2 | 2,4-di*tert*-butylphenol | 206.32 | 2.13 | $C_{14}H_{22}O$ | 191, 57, 206 | Aromatic hydrocarbon | Anti-inflammatory, antimicrobial and antioxidant *USDA, 1992*–2016 [U.S. Department of Agriculture, Agricultural Research Service]). Dr. Duke's) |
| 3 | pentacosane | 352.68 | 1.66 | $C_{25}H_{52}$ | 57, 71, 85 | Alkane | Antimicrobial and antioxidant (*Marrufo et al., 2013*) |
| 4 | henicosane | 296.57 | 0.96 | $C_{21}H_{44}$ | 85,71,57 | Alkane | Antiasthmatics, urine acidifiers and antimicrobial (*Usha Nandhini, Sangareshwari & Lata, 2015*) |
| 5 | triacontane | 422.81 | 1.4 | $C_{30}H_{62}$ | 57, 85, 113 | Alkane | Antimicrobial and cytotoxic (*Hsouna et al., 2011*), antidiabetic, antitumor and antibacterial (*Tiagy & Agarwal, 2017*) |
| 6 | hentriacontane | 436.83 | 1.63 | $C_{31}H_{64}$ | 57, 85, 113 | Alkane | Antibacterial activity (*Olubunmi et al., 2009*), and anti-inflammatory (*Kim et al., 2011*) |
| 7 | octacosane | 394.76 | 1.28 | $C_{28}H_{58}$ | 57, 141, 239 | Alkane | Antimicrobial and antioxidant (*Jun et al., 2018*) |

phytochemical compounds with biological activities previously reported (Tables 6 and 7); their mass spectra resulting from the GC-MS analyses and chemical structures are presented in Figs. S1–S4.

Ten phytochemical compounds shown in Fig. 3 were the most prevailing in the two extracts (CHCl$_3$ and MeOH) and both membrane filters (PVDF and PTFE): Benzene, 1,3-bis(1,1-dimethylethyl) (4.19%) in MeOH/PVDF and (3.53%) in (MeOH/PTFE), hexadecanoic acid (4.67%) in MeOH/PVDF; 1,3-benzothiazole (5.33%), butyl hexadecanoate (4.83%), 2-methylpropyl octadecanoate (3.10%), (6E,10E,14E,18E)-2,6,10,14,18-pentamethylicosa-2,6,10,14,18-pentaene (4.80%), and 17-(5-ethyl-6-methylheptan-2-yl)-10,13-dimethyl-2,7,8,9,11,12,14,15,16,17-decahydro-1H-cyclopenta[a]phenanthrene (13.80%) in CHCl$_3$/PVDF; and 4-ethyl-1,2-dimethylbenzene (4.24%), 2,4-di*tert*-butylphenol (5.28%) and 2-(2-ethylhexoxycarbonyl)benzoic acid (7.55%) in CHCl$_3$/PTFE (Fig. 3).

**Table 7 Phytochemical compounds with biological activity identified by GC-MS in CHCl₃ extract.**

| Peak No. | Name of the compound | Molecular weight (g/mol) | Peak area (%) | Molecular formula | Ions (m/z) | Compound nature | Biological activity |
|---|---|---|---|---|---|---|---|
| **PVDF filter** | | | | | | | |
| 1 | 1,3-benzothiazole | 135.18 | 5.33 | $C_7H_5NS$ | 135, 108, 69 | Aromatic Hydrocarbon | Anticonvulsant, anti-inflammatory, antileishmanial, antimicrobial and antitumor (*Siddiqui, Khan & Rana, 2007*) |
| 2 | (E,7R,11R)-3,7,11,15-tetramethylhexadec-2-en-1-ol | 296.53 | 0.45 | $C_{20}H_{40}O$ | 55, 71, 81 | Alcohol | Antispasmodic (*Pongprayoon et al., 1992*), anticarcinogen *USDA, 1992*–2016 [U.S. Department of Agriculture, Agricultural Research Service]). Dr. Duke's; *Lee, Lee & Park, 1999*; *Hema, Kumaravel & Alagusundaram, 2011*), antitubercular (*Saikia et al., 2010*), antibacterial, antifungal, antimalaria, analgesic and stimulant (*Hema, Kumaravel & Alagusundaram, 2011*), anticonvulsant (*Costa et al., 2012*), anti-inflammatory, antinociceptive (*Okiei et al., 2009*; *Silva et al., 2014*; *Islam et al., 2018*), anxiolytic, cell autophagy and apoptosis inducer metabolism-modulating, cytotoxic and immune-modulating (*Islam et al., 2018*), antimicrobial (*Islam et al., 2018*), and antioxidant (*Mohammad, Omran & Hussein, 2016*; *Islam et al., 2018*) |
| 3 | 7,9-di*tert*-butyl-1-oxaspiro[4.5]deca-6,9-diene-2,8-dione | 276.37 | 1.61 | $C_{17}H_{24}O_3$ | 57, 175, 217 | Ketone | Antioxidant (*USDA, 1992*–2016 [U.S. Department of Agriculture, Agricultural Research Service]). Dr. Duke's) |
| 4 | methyl hexadecanoate | 270.45 | 1.1 | $C_{17}H_{34}O_2$ | 74, 143, 227 74, 87, 43, 55, 143 | Fatty Acid | Antibacterial (*Agoramoorthy et al., 2007*), antifungal, anti-inflammatory, blood cholesterol decrease (*Hema, Kumaravel & Alagusundaram, 2011*), antioxidant (*Agoramoorthy et al., 2007*; *Hema, Kumaravel & Alagusundaram, 2011*) |
| 5 | methyl octadecanoate | 298.50 | 0.97 | $C_{19}H_{38}O_2$ | 143, 74, 55 | Fatty Acid | Antimicrobial (*Abubakar & Majinda, 2016*) |
| 6 | butyl hexadecanoate | 312.53 | 4.83 | $C_{20}H_{40}O_2$ | 257, 129, 56 | Fatty Acid | Antioxidant (*Prakash, Gondwal & Pant, 2011*), and antimicrobial (*Sujatha et al., 2014*) |
| 7 | icosane | 282 | 1.34 | $C_{20}H_{42}$ | 113, 85, 57 | Alkane | Antibacterial (*Boussaada et al., 2008*; *Hsouna et al., 2011*), antifungal, antitumor and cytotoxic (*Hsouna et al., 2011*) |
| **PTFE filter** | | | | | | | |
| 1 | 2,4-di*tert*-butylphenol | 206.32 | 5.28 | $C_{14}H_{22}O$ | 57, 191, 206 | Aromatic hydrocarbon | Anti-inflammatory, antimicrobial and antioxidant *USDA, 1992*–2016 [U.S. Department of Agriculture, Agricultural Research Service]). Dr. Duke's) |
| 2 | 2-(2-ethylhexoxycarbonyl)benzoic acid | 278.34 | 7.55 | $C_{16}H_{22}O_4$ | 149, 167, 112 | Ester | Cytotoxic (*Krishnan, Mani & Jasmine, 2014*) |

Identity from five compounds that showed a similarity percentage above 95%, was supported by comparison of their retention times with pure commercial standards (Figs. S5–S9). 1,3-benzothiazole was found at RT of 14.61 min, with ions (m/z) of 135 and

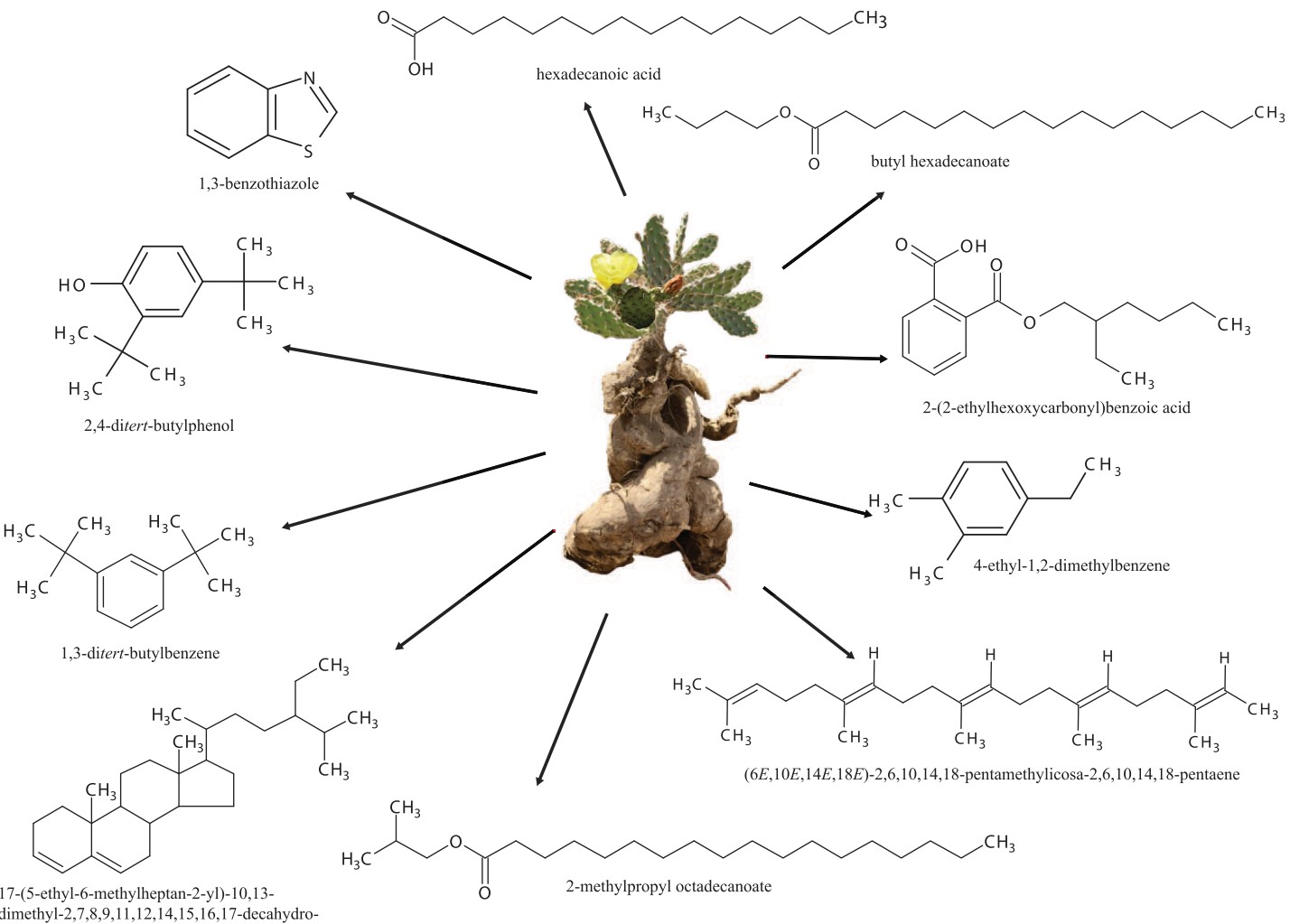

**Figure 3 Chemical structures.** Prevailing phytochemical compounds of *Opuntia megarrhiza* identified in GC-MS analysis in CHCl₃ and MeOH extracts with PVDF and PTFE membrane filters.

107.9. henicosane at RT of 33.5 min, with ions (*m/z*) of 57, 113. hentriacontane at RT of 48.7 min, with ions (*m/z*) of 57, 85, and 113. methyl hexadecanoate at RT of 30.5 min, with ions (*m/z*) of 74, 143, 227, and 55. triacontane was detected at RT of 33.5 min, with ions (*m/z*) 57, 85 and 113.

Finally, 34 compounds with no identified biological activity were found, eight in the (CH₃)₂CO/MeOH extract with PVDF membrane filter (MeOH/PVDF), five in the (CH₃)₂CO/MeOH extract with PTFE (MeOH/PTFE), 16 in the (CH₃)₂CO/CHCl₃ extract with PVDF (CHCl₃/PVDF), and five in the (CH₃)₂CO/CHCl₃ extract with PTFE (CHCl₃/PTFE).

## DISCUSSION

The use of *Opuntia megarrhiza* in traditional medicine in Mexico has been reported previously (*Segura-Venegas & Rendón-Aguilar, 2016*), however, this is the first study that

demonstrate the presence of phytochemical compounds with biological activities. *Opuntia* species are used in the world as local medicinal interventions for chronic diseases and as food sources, mainly because they possess nutritional properties and biological activities that has been recently reviewed (*Aruwa, Amoo & Kudanga, 2018*). Here we report for the first time, the identification of several phytochemical compounds in *O. megarrhiza* with biological activities. Our findings highlight the relevance of this species in developing of new drugs, trough future chemical studies, and encourage of planting this species once this one is listed as endangered in the IUCN Red List.

Biotechnological methods are reliable and provide continuous sources of raw material and natural products for food, pharmaceutical, and cosmetic industries (*Rao & Ravishankar, 2002*; *Nalawade et al., 2003*; *Julsing, Quax & Kayser, 2007*). Previously, it has been indicated that more than 50,000 plant species are used in phytotherapy and medicine, and around 66% of them are harvested from nature leading to local extinction of many species or degradation of their habitats (*Tasheva & Kosturkova, 2012*). Alternatives to protect these useful plants, should be directed to both preservation of the plant populations and elevating the level of knowledge for sustainable utilization of these plants in medicine have been previously indicated (WHO, 2010, http://www.who.int/mediacentre/factsheets/fs134/en/). Biotechnological methods offer possibilities not only for faster cloning and conservation of the genotype of the plants (*Verpoorte, Contin & Memelink, 2002*; *Tripathi & Tripathi, 2003*) but for modification of their gene information, regulation, and expression for production of valuable substances in higher amounts or with better properties (*Rao & Ravishankar, 2002*; *Khan et al., 2009*).

GC-MS is one of the most precise methods to identify various metabolites present in plant extracts (*Fiehn et al., 2000*; *Roessner et al., 2000*; *Roessner et al., 2001*; *Kopka, 2006a*; *Kopka, 2006b*; *Fiehn, 2006*; *Fernie, 2007*; *Saito & Matsuda, 2010*; *Tiago et al., 2016*; *Dinesh-Kumar & Rajakumar, 2018*) since some of these chromatographs include preloaded libraries or databases (NIST and WILEY) that allows to know the possible identity of the metabolites by comparing the resulting mass spectra with those found as reference in these libraries (*Kim et al., 2019*; *Wei et al., 2014*). Several studies indicate that *Opuntia* plants contain different phytochemical groups such as phenolic acids, sterols, esters, coumarins, terpenoids, and alkaloids with several health benefits (*Piattelli, Minale & Prota, 1965*; *Stintzing, Schieber & Carle, 2001*; *Strack, Vogt & Schliemann, 2003*; *Paiz et al., 2010*; *Osorio-Esquivel et al., 2011*; *Aruwa, Amoo & Kudanga, 2018*). However, the nature of the compound extracted depends largely on their solubility in the extraction solvent, the degree of polymerization of the phenols, and the interaction of the phenols with other constituents of the plant (*Choi et al., 2002*; *El Cadi et al., 2020*). But the use of different membrane filters allows to identified chemical compounds with different hydrophobicity and molecule sizes. Previously, it has been indicated that PTFE has less hydrophobic adsorption but more size exclusion (*Xiao et al., 2014*).

In addition, identity of five of the compounds found was corroborated using pure commercial standards. The ions obtained from each of the standards corresponded to those found in the extracts according to the NIST base of the equipment. GC-MS has a library of mass spectra, which makes it easy to obtain compounds that have the most

similar mass to the library spectrum (*Kim et al., 2019*). However, the attribution of a GC-MS chromatographic peak should be confirmed whenever possible by comparison with a standard compound analyzed under the same experimental conditions (*Sturaro, Parvoli & Doretti, 1994*). We identified five compounds in the extracts performed through the use of standards. In this context, the analytical standard is used as a reference in the qualitative, quantitative and identity determinations of a compound, it must also have high purity and stability (*Sun et al., 2015*).

On the other hand, the major phytochemical compounds found in our study have been reported to possess several biological activities. Some alkanes like hentriacontane and triacontane have antibacterial activity (*Boussaada et al., 2008*; *Olubunmi et al., 2009*; *Hsouna et al., 2011*; *Tiagy & Agarwal, 2017*). Heptadecane have antifungal activity (*Adeleye, Daniels & Omadime, 2010*). Icosane has both antibacterial and antifungal activity (*Hsouna et al., 2011*). Henicosane, heptadecane, octacosane, and pentacosane have antimicrobial activity (*Rahbar, Shafagha & Salimi, 2012*; *Marrufo et al., 2013*; *Usha Nandhini, Sangareshwari & Lata, 2015*; *Jun et al., 2018*). Heptadecane and hentriacontane have anti-inflammatory activity (*Kim et al., 2011*; *Kim et al., 2013*). Heptacosane, heptadecane, octacosane, and pentacosane have antioxidant activity (*Kim et al., 2013*; *Marrufo et al., 2013*; *Jun et al., 2018*). Icosane and triacontane have antitumor activity (*Hsouna et al., 2011*; *Tiagy & Agarwal, 2017*). Triacontane has antidiabetic activity (*Tiagy & Agarwal, 2017*). Henicosane has been reported as an antiasthmatic, urine acidifier (*Usha Nandhini, Sangareshwari & Lata, 2015*). Icosane, octadecane, and hexadecanoic acid has been previously identified in *Opuntia stricta* (*Izuegbuna, Otunola & Bradley, 2019*).

Fatty acids like octadecanoic acid have antibacterial or antifungal activity (*Gehan et al., 2009*; *Hsouna et al., 2011*), and it has previously identified in *Opuntia dillenii* (*Ben-Lataief et al., 2020*). The hexadecanoic acid have antialopecic, anti-androgenic, antifibrinolytic, antioxidant, antipsychotic, hemolytic, hypocholesterolemic, nematicide, pesticide, 5-Alpha reductase inhibitor (*USDA, 1992*–2016 [U.S. Department of Agriculture, Agricultural Research Service]), and anti-inflammatory (*Aparna et al., 2012*). Octadecanoic acid have been reported as anticarcinogen or antitumoral (*Hsouna et al., 2011*; *Gehan et al., 2009*). Fatty acid like butyl hexadecanoate, methyl hexadecanoate, and methyl octadecanoate, has antimicrobial activity (*Sujatha et al., 2014*; *Abubakar & Majinda, 2016*). Benzenoids like 2-(2-ethylhexoxycarbonyl)benzoic acid ester has been reported as cytotoxic (*Krishnan, Mani & Jasmine, 2014*). The diterpene (*E*,7*R*,11*R*)-3,7,11,15-tetramethylhexadec-2-en-1-ol has been reported to have multiple activities like anticarcinogen, anticonvulsant, antifungal, anti-inflammatory, antimalaria, antimicrobial, antinociceptive, antioxidant, antitubercular, antispasmodic, anxiolytic, autophagy and apoptosis inducing, cytotoxic, immune-modulating, metabolism-modulating, resistant to gonorrhea, and stimulant (*USDA, 1992*–2016 [U.S. Department of Agriculture, Agricultural Research Service]; *Pongprayoon et al., 1992*; *Lee, Lee & Park, 1999*; *Okiei et al., 2009*; *Saikia et al., 2010*; *Hema, Kumaravel & Alagusundaram, 2011*; *Costa et al., 2012*; *Silva et al., 2014*; *Mohammad, Omran & Hussein, 2016*; *Islam et al., 2018*), and the 2-(2-ethylhexoxycarbonyl)benzoic

acid has anti-inflammatory, antimicrobial, antioxidant, antiviral, and cytotoxicity activities (*Krishnan, Mani & Jasmine, 2014*).

Additionally, phytochemical compounds we found in *Opuntia megarrhiza* with no reports of biological activity, have been previously identified in other *Opuntia* species. For example, (*Z*)-octadec-9-enoic acid and 17-(5-ethyl-6-methylheptan-2-yl)-10,13-dimethyl-2,7,8,9,11,12,14,15,16,17-decahydro-1*H*-cyclopenta[a]phenanthrene was identified in *O. dillenii* (*Ben-Lataief et al., 2020*). Additionally, β-Sitosterol is the major sterol extracted from different parts of the fruit oils of *Opuntia ficus-indica* (*Ramadan & Mörsel, 2003a*, *2003b*). Herein, we identify (3*S*,8*S*,9*S*,10*R*,13*R*,14*S*,17*R*)-17-[(2*R*,5*S*)-5-ethyl-6-methylheptan-2-yl]-10,13-dimethyl-2,3,4,7,8,9,11,12,14,15,16,17-dodecahydro-1*H*-cyclopenta[a]phenanthren-3-ol in *O. megarrhiza*.

## CONCLUSIONS

The GC-MS analysis of cladode extracts of *Opuntia megarrhiza* conducted here proves the presence of several phytochemical compounds responsible for biological activities previously reported support the medicinal use of this plant in traditional medicine. In particular, the anti-inflammatory activity in compounds with a high similarity percentage in our results (*e.g.*, hexadecanoic acid, 2,4-di*tert*-butylphenol, hentriacontane, 1,3-benzothiazole, and methyl hexadecanoate) supports its use for treating bone fractures. Hence, *O. megarrhiza* represents a source for finding phytochemical compounds with potential use in medicine and biotechnology. Our results represent an advance in the knowledge of an endangered plant, not previously studied, and with ethnical uses, and support future target studies through the identification of compounds with biotechnological potential using certified standard and additional tools.

### Funding

Madeleyne Cupido received a grant of CONACYT (Graduate Studies Scholarship 1007054). This research was funded by the international cooperative research of Rural Development Administration (RDA) from Republic of Korea, (Grant PJ012429012016 to Pablo Delgado-Sánchez) and CONACyT (Grant 2014-243454 to JADN). The funders had no role in study design, data collection and analysis, decision to publish, or preparation of the manuscript.

### Grant Disclosures

The following grant information was disclosed by the authors:
CONACYT: 1007054.
International Cooperative Research of Rural Development Administration (RDA), Republic of Korea: PJ012429012016.
CONACyT: 2014-243454.
## Competing Interests

The authors declare that they have no competing interests.

## Author Contributions

- Madeleyne Cupido conceived and designed the experiments, performed the experiments, analyzed the data, performed the computation work, prepared figures and/or tables, authored or reviewed drafts of the paper, and approved the final draft.
- Arturo De-Nova conceived and designed the experiments, analyzed the data, performed the computation work, prepared figures and/or tables, authored or reviewed drafts of the paper, contributed materials and reagents, and approved the final draft.
- María L. Guerrero-González conceived and designed the experiments, analyzed the data, prepared figures and/or tables, authored or reviewed drafts of the paper, contributed analysis tools, and approved the final draft.
- Francisco Javier Pérez-Vázquez conceived and designed the experiments, performed the experiments, analyzed the data, performed the computation work, prepared figures and/or tables, authored or reviewed drafts of the paper, contributed materials and reagents, and approved the final draft.
- Karen Beatriz Méndez-Rodríguez performed the experiments, analyzed the data, prepared figures and/or tables, authored or reviewed drafts of the paper, and approved the final draft.
- Pablo Delgado-Sánchez conceived and designed the experiments, analyzed the data, performed the computation work, prepared figures and/or tables, authored or reviewed drafts of the paper, contributed materials and reagents, and approved the final draft.

## Data Availability

The raw data is available in the Supplemental File.

## Supplemental Information

Supplemental information for this article can be found online at http://dx.doi.org/10.7717/peerj-ochem.5#supplemental-information.

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
