# Peer review of "GC-MS analysis of phytochemical compounds of *Opuntia megarrhiza* (Cactaceae), an endangered plant of Mexico"

_PeerJ Organic Chemistry, doi:10.7717/peerj-ochem.5_

## Round 0.1 · original submission · Major Revisions

I have received the comments of three experts in natural compounds analysis. As you can see by the reviews, your work needs considerable adjustments to be acceptable for publication. However, if you are able to address the questions and suggestions of each reviewer (mainly reviewers #1 and #3) I could reconsider my decision.

Reviewer 1 ·

Basic reporting

In the manuscript entitled as GC-MS analysis of phytochemical compounds of Opuntia megarrhiza (Cactaceae), an endangered plant of Mexico the authors describe the identification and characterization of the phytochemical compounds present in wild individuals of O. megarrhiza using Gas Chromatography Coupled to Mass Spectrophotometry. In this study, the authors reported 53 phytochemical compounds, 18 have previously identified with some biological activity. The variety of metabolites presents in O. megarrhiza justifies the medicinal use of this plant in traditional medicine and highlight it as a source of phytochemical compounds with potential in medicine and biotechnology.
The journal PeerJ Organic Chemistry publishes articles exploring subjects including biochemistry, polymers, green chemistry, photochemistry, and synthetic organic chemistry. I believe that the present study is in accordance with the profile of the journal, that the manuscript is relatively free of error and demonstrates a quality of the paper in terms of scientific value. Therefore, I recommend this manuscript for publication in PeerJ Organic Chemistry after consideration of the issues and corrections.

1) The whole manuscript is clearly written in professional, unambiguous language. The introduction section is full of information about the phytochemical compounds, however, I think you should do an update in some references as at lines 37-43, to show the topic is more current. Some examples: Sci Rep 11, (2021) 10041 (https://doi.org/10.1038/s41598-021-89437-4); Food Chemistry 362 (2021) 130196 (https://doi.org/10.1016/j.foodchem.2021.130196); Biotechnology Reports 30 (2021) e00633 (https://doi.org/10.1016/j.btre.2021.e00633).

2) I strongly recommend that the authors draw the chemical structures to the demonstrate the phytochemical compounds with biological activities (in the Tables 6 and 7). In this Tables (6 and 7) you should exclude the Similarity (%) and RT (min) columns, they are a duplicate information.

Experimental design

1) Some studies in current literature report the use of LC-MS–MS to identification and characterization of components in plant extracts. With this kind of Cactaceae plant, could the authors use this technique? Or is the GC-MS the most appropriate for this situation?

2) The method was described with sufficient detail and information to replicate. However, the authors need to correct some abbreviations of the solvents. Methanol is correctly represented as MeOH, not MeOTH. Still, acetone isn’t CH₃CH₃, this is ethane molecule. The correct acetone representation is (CH3)2CO.

3) In some point of the manuscript (line 159, 172 and go on) the authors starting described the phytochemical constituents from the CH3CN extract. Is this information correct? The use of acetonitrile wasn’t described in the Materials and Methods section.

Validity of the findings

1) Only the compounds and their respective activities were reported in the Discussion section, and the obtained results support the medicinal use of this plant in traditional medicine. However, the authors should explore more the importance of why find out what kind of phytochemical compounds are present in O. megarrhiza. For example, you should discuss about a future developing of new drugs, new chemical studies, or about the encouragement of planting this species once this one is listed as endangered in the Red List of the IUCN.

2) In the line 198 the authors described the “the presence of 18 phytochemical compounds with biological activities, with similarity values of identification above 93%.” Considering the tables 1-4, there are only 15 phytochemical compounds with similarity values of identification above 93%. Is this count correct? Still, a short sentence about the similarity values of identification and if there are a percentage acceptable to characterize a molecule, should be add in the discussion.

3) For better molecular characterization, the MS spectrum of compounds at the Table 6 and 7, should be add in the Supplemental files.

Additional comments

Please correct following typos and errors:
- change from 5ºC to 5 ºC (line 129)
- change from 300ºC to 300 ºC (line 129)
- the number 3 in the compound is subscript (e.g. CHCl3) (lines 144, 159, 172, 188 and go on)
- the letter H in 2H-Pyran-2-one (line 162) is in italic form.
- change from MetOH3 to MeOH (lines 185, 186, 192 and go on, Figure 3, Table 5)
- change from CHCl to CHCl3 (line 193)
- the quality of Figure 2 should be improved.

Reviewer 2 ·

Basic reporting

The MS “GC-MS analysis of phytochemical compounds of Opuntia megarrhiza (Cactaceae), an endangered plant of Mexico”, authored by Arturo De-Nova and Pablo Delgado-Sánchez et al., describes the identification and characterize the phytochemical compounds present in individuals of O. megarrhiza using the gas chromatography coupled to mass spectrometry. The authors used organic solvents to prepare the extracts and 53 phytochemical compounds were identified. The protocol used in the chemical characterization agrees with literature procedures. However, the text is not clear and need to an extensive revision for types. The quality of figures and chemical structures also need to be improved. After reading of the manuscript, it is possible to identify a confusion in the nomenclature/abbreviation of the solvents used to prepare the extracts, which makes it difficult to understand the text. This work could be a relevant contribution for Chemistry of Natural Products, but I don’t recommend the acceptance of this manuscript in Peer J. this way. I suggest a careful review of the text and addition of a biological application study for the extracts before a new submission.

Experimental design

The research agrees with the Scope of the journal. However, in some cases the methods are not described with sufficient details or information to replicate. I suggest a detail review.

Validity of the findings

The results obtained are promising. However, they need to be better discussed. I suggest adding another reference parameter to complete the characterization of the compounds present in the extracts since the similarity of the spectra in many cases is less than 75%. This comment applies to the results described in tables.

·

Basic reporting

While the work is interesting and relevant regarding the phytochemical analysis of Opuntia megarrhiza, I have some concerns about the manuscript that make me not to recommend publication in its current form. Thus, it needs major revisions before publication, including English language improvement.

Experimental design

In my opinion, the main concern is the tentative identification of the extracted phytochemicals based only on the similarity approach from the Mass Spectral Libraries. Although the libraries are an important tool to help in the identification, additionally analysis should be used. At least, I recommend a Linear retention index analysis and comparison with standard samples.

Validity of the findings

The authors mention that the extracts were filtered using two different membrane filter (PTFE and PVDF) and as consequence, the chemical composition of the extracts are different. Thus, the authors should clarify the difference between both membranes and discuss the results regarding the composition.

Regarding the chromatograms showed in the Figure 2, some of the labeled peaks are nearly in the noise, which difficult the appropriated identification of the phytochemicals.
I also suggest improving the discussion comparing the identified compounds and the identified compounds reported in other studies for the Opuntia Species

The authors reposted that a” exhaustive search” was made for the biological activity based in the literature. But, if possible, a small biological activity screening against bacteria or fungi will considerably improve the study. As suggestion too, due to the large number of cited references in the manuscript (more the 100 references) a revision manuscript about the phytochemical composition and biological activity from Opuntia Species could be organized.

Additional comments

Some minor revision to improve the manuscript are suggested as follow:
Mass analysis is a spectrometric technique and not a ”Spectrophotometry” technique. Also, use the word “technique” instead of the word “technic”.
In the Abstract, the sentence “and they were analyzed in a mass spectrometry detector with eletronic impact” could be replaced with “and they were analyzed by gas chromatography-electron impact-mass spectrometry”.
The introduction should be revised. It is very repetitive.
In line 30, the sentence seems incomplete. “127 general…”
In line 71, the authors should replace de “UV resonance“, with “UV-Vis spectrophotometry”. UV-Vis spectrophotometry is not a resonance spectroscopy.
Line 113: correct “past” to “paste”.
Line 114: The for methanol is MeOH and not MeOTH or MetOH3 as shown in Table 5. Check all the acronyms along the manuscript. Also, check all the molecular formulas in the mains text and Tables, such as for acetone which represented as CH3CH3
Line 132: Correct the misspelt word “specters”
All the compounds nomenclature should be revised.
The symbol for mass/charge (m/z) is always italicized. Check it in the main manuscript and Tables

---

## Round 0.2 · Major Revisions

I have received the comments of two reviewers. As you can see, according one of them, the manuscript still needs considerable adjustments. I'd like to ask you to address the questions and suggestions of reviewer #3) for further evaluation.

Reviewer 1 ·

Basic reporting

no comment

Experimental design

no comment

Validity of the findings

no comment

Additional comments

In the manuscript entitled as “GC-MS analysis of phytochemical compounds of Opuntia megarrhiza (Cactaceae), an endangered plant of Mexico” the authors described the identification and characterization of the phytochemical compounds present in wild individuals of O. megarrhiza using Gas Chromatography Coupled to Mass Spectrometry. In this study, the authors reported 53 phytochemical compounds, 19 have previously identified with some biological activity. The variety of metabolites presents in O. megarrhiza justifies the medicinal use of this plant in traditional medicine and highlight it as a source of phytochemical compounds with potential in medicine and biotechnology.
I believe that the present study is in accordance with the profile of the journal, that the manuscript is well written and after all changes performed by the authors, by adding current literature, improve de discussion, and included the chemical structures and MS spectrum of compounds in Supplementary, I have seen an improvement in the quality of the manuscript. Thus, I consider this article accepted for publication in PeerJ Organic Chemistry.

·

Basic reporting

The author has substantially improved the manuscript and incorporated major of the reviewers' suggestions. However, regarding the extracts identification, the identity of five compounds was tentatively confirmed by correlation with standards. However, the remaining 48 compounds were identified based only on the mass spectra comparison. With the possibility to easily apply the Linear retention index analysis, I keep the suggestion to use it.
More important, even with the use of the pure standards, the mass spectra for the standard benzothiazole (Figure S5-A) are not comparable to the spectra of the benzothiazole in the extract (Figure S3-A). The fragmentation patterns are very different. The same occurs to the heneicosane. In addition, for a better comparison, the retention times should be informed in the standard cromatograph spectra.
The spectra presentation and analysis still is the main concern and requires attention.

Experimental design

no comment

Validity of the findings

no comment

Additional comments

In addition, it is still possible to identify some typos.
Line 121. I suggest replacing the expression “These solutions were mixed by centrifuge one hour” with “These solutions were vortex mixed one hour “. I believe that in this case, a centrifuge was not used once it accelerated particle sedimentation. Thus, this procedure has to be checked.
The nomenclature of the compounds still requires revision. The library software has some limitations regarding to the way how the nomenclature is displayed. Thus, in the main manuscript, the prefix such cis or trans, (Z) or (E) has to be italicized. Also, the prefix (R)-, (S)-, n-, tert-. The nomenclature en the Tables also has to be revised.
In this way, I strongly suggest revising de nomenclature of the compounds following the IUPAC rules. Otherwise, I suggest including in the experimental Section that the Chemical Abstract nomenclature is used.
Line 208: CHCl3 and not CHCL3. In the line 2019 and 220 also.
Line 213: For retention time, I suggest using tR or RT instead of Tr. The abbreviation for minute is “min” and does not require de dot.
Line 217, 218, and 219. It is missing the open parenthesis in the acetone formula.
In the Figure 2 caption, the representation of methanol must be corrected. MeOH and not MeOTH. In addition, the Figure 2 caption mentions the CH3CN extract. According to the main text, it should be chloroform (CHCl3) and not acetonitrile. It has to be checked.
Figure 3. replace MeOTH with MeOH, and the use or not of CH3CN has to be checked.
In Figure S2, the chemical structure shown in the spectra (B) does not represent the Phenol, 2,4-bis(1,1-dimethylethyl-), but its silyl ether derivative. It is also observed in Figure S4, spectrum A. Please revise it.
In Figure S5, the spectrum F comes after spectrum D. The retention times for each standard should be informed for comparison with the extract chromatograms.
Also, the mass spectra for the standard benzothiazole (Figure S5-A) are not comparable to the spectra of the benzothiazole in the extract (Figure S2-A).

---

## Round 0.3 · accepted · Accept

Dear Dr Delgado-Sánchez

The new version of your manuscript was revised by reviewer #3 and myself.

I'm happy to inform you that your manuscript has been accepted for publication in PeerJ Organic Chemistry.

Thanks for your submission.

·

Basic reporting

no comment

Experimental design

no comment

Validity of the findings

no comment

Additional comments

The major revisions request were considered by the authors and now the manuscript is suitable for publication.